

# A study on an efficient citrus Huanglong disease detection algorithm based on three-channel aggregated attention

Yizong Wang, Zhengrong Xiao, Hong Wang, Fei Li and Jiya Tian

School of Information Engineering, Xinjiang Institute of Technology, Aksu, Xinjiang, China

## ABSTRACT

**Background:** Aiming at the problems of complex and diverse field symptoms of citrus Huanglong disease (HLB), low efficiency and insufficient recognition accuracy of traditional detection methods, this study proposes an efficient detection algorithm based on improved You Only Look Once (YOLO)v8.

**Methods:** Firstly, a new character to float (C2f) Attention inverse residual moving block (IRMB) module is designed, which significantly enhances the model's sensitivity to tiny disease features while reducing the number of parameters by fusing the lightweight IRMB with the adaptive attention gating mechanism, and solves the problem of losing key texture information due to downsampling in the traditional C2f module. Secondly, the three-channel aggregated attention module Powerneck is proposed in the Neck section, which realizes efficient cross-scale feature interactions, effectively suppresses background noise interference, and improves robustness in complex field scenes through SimFusion_4in feature alignment, information fusion module (IFM) global context fusion, and Power channel dynamic weighting strategy. In addition, the detection head design is optimized by structural reparameterization technique to further accelerate the inference process.

**Results:** The experimental results show that on the citrus dataset containing 12 diseases and two health states, the mAP50 of this model reaches 97% and the accuracy is 91.5%, which is 1.1% and 1.2% higher than that of the original YOLOv8, respectively, and the inference speed is improved by 14.6% to 370 frames per second (FPS). Comparison of the different models shows that the C2f Attention IRMB, through the mechanism of dual attention The comparison of different models shows that C2f Attention IRMB strengthens the feature expression ability through the dual-attention mechanism, and the Powerneck module reduces redundant computation through dynamic channel pruning, and the two synergistically optimize the model performance significantly. Compared with mainstream models such as YOLOv5m and YOLOv7x, this method is more advantageous in the balance of accuracy and speed, and can meet the demand of real-time detection in the field.

**Discussion:** The algorithm provides an efficient tool for early and accurate identification of citrus Huanglong disease, which is of great practical significance for reducing pesticide misuse and improving the efficiency of orchard management, and also provides new ideas for the design of lightweight target detection models in agricultural scenarios.

Corresponding author
Hong Wang, 891775971@qq.com

## INTRODUCTION

Mandarin oranges, one of China's most productive fruits, boast an annual standing volume of approximately 6,039,000 tons, underscoring their significant economic value. The pulp is not only processed into juice, delivering essential nutrients to the human body but the rind is also utilized as a spice and fragrance, enjoying considerable popularity among consumers. Furthermore, citrus fruits possess substantial medicinal properties and can be employed in the treatment of various diseases (*Wang et al., 2023*). As a significant commercial fruit, the quality of citrus plays a crucial role in the development of the citrus industry. One of the most critical factors influencing citrus quality is the array of diseases to which the fruit may be susceptible throughout its growth. Implementing effective disease management strategies is essential to minimize losses in the orchard (*Zhang et al., 2024b*). Citrus Huanglong disease presents a significant threat to the development of China's citrus industry due to its complex and varied symptoms, which severely impact both yield and quality. This disease is caused by the citrus Huanglong disease bacterium, leading to prominent symptoms such as leaf yellowing, wilting, and the appearance of yellow spots on fruit surfaces. These manifestations significantly diminish the yield and quality of affected citrus crops. Traditional diagnostic methods for citrus Huanglong disease include field diagnosis, electron microscopy, and hyperspectral imaging; however, each of these approaches has inherent limitations, such as high detection costs, complex operational procedures, and limited adaptability. Consequently, these limitations result in an inefficient detection process with variable accuracy. More critically, such diagnostic services are typically confined to regions that possess the necessary manpower and infrastructure to support them (*Zhang, Xun & Chen, 2022*). In recent years, the rapid advancement of computer vision technology has made deep learning-based algorithms for detecting field symptoms of citrus Huanglong disease a focal point of research. Addressing the challenges posed by small sample sizes, uneven distribution, and the difficulty of collecting data on citrus leaf diseases, *Dai et al. (2023)* implemented an innovative enhancement based on FastGAN2 and EfficientNet-B5. The results demonstrated that the modified model achieved improvements of 2.26%, 1.19%, 1.98%, and 1.86% in accuracy, precision, recall, and F1 score, respectively, when compared to the original model. However, it is important to note that this method serves primarily as an effective tool for the image classification of citrus diseases and nutrient deficiency disorders with limited sample sizes (*Dai et al., 2023*). *Syed-Ab-Rahman, Hesamian & Prasad (2021)* developed a two-stage deep convolutional neural network (CNN) model, proposing an end-to-end, anchor frame-based deep learning approach for the detection and classification of citrus diseases. The results of their study demonstrated that the model could effectively identify and differentiate among three distinct citrus diseases, highlighting its potential as a valuable decision-support tool for growers and farmers in the identification and

classification of citrus diseases (*Syed-Ab-Rahman, Hesamian & Prasad, 2021*). *Yadav et al. (2022)* developed an innovative CNN-based framework for the detection and classification of citrus diseases. The findings of this study demonstrate that the model has significant decision-making implications for citrus growers by enabling them to effectively recognize and classify various citrus diseases (*Yadav et al., 2022*). *Dananjayan et al. (2022)* employed a radial basis function support vector machine (RBF SVM) in conjunction with a customized shallow CNN to classify citrus diseases, achieving remarkable performance with an overall accuracy of 93%. This highlights the significance of hyperspectral imaging systems for the automated classification of diseases affecting citrus fruits and leaves (*Dananjayan et al., 2022*). By combining fluorescence imaging spectroscopy with a supervised learning approach, *Neves et al. (2023)* successfully identified diseases such as citrus rot, achieving an accuracy of up to 95%. Compared to traditional polymerase chain reaction-based methods, this approach resulted in significant time and cost savings (*Neves et al., 2023*). In a different study, *Zhang et al. (2024a)* introduced a frequency domain attention network (FdaNet) that adaptively adjusts the weights of each frequency based on the importance of feature information across different frequency domains during the inference process. To address the complexity and diversity of citrus disease data, they also proposed a hybrid attention network (HaNet) focused on multidimensional feature information. Experimental results from their study showed that the proposed model attained recognition rates of 98.83% and 98.77% on 50-layer and 101-layer networks, respectively, demonstrating excellent performance (*Zhang et al., 2024a*). Addressing challenges such as large model sizes, slow detection rates, and insufficient accuracy in current target detection algorithms, *Li et al. (2024)* developed a lightweight detection method for citrus leaf disease based on an improved Single Shot MultiBox Detector (SSD). In comparison to Visual Geometry Group (VGG)-16-SSD, the average accuracy of this improved model increased by 4.4%, enabling rapid and accurate diagnosis of citrus leaf diseases, which is crucial for timely and precise medical applications at the disease site (*Li et al., 2024*). *Holkar et al. (2024)*, *Kunduracioglu & Pacal (2024)*, and *Pandey & Abhimanyu (2023)* effectively classified and diagnosed grape leaf diseases by fine-tuning an advanced pre-trained CNN and visual transformer model, providing a powerful tool for enhancing crop yields and grape variety identification, with the potential for widespread agricultural application. The lightweight algorithm proposed by *Gao et al. (2024)* improves the You Only Look Once (YOLO)v8n framework, enhancing the accuracy and efficiency of disease detection through techniques such as GhostConv and a global attention mechanism. Experimental evidence indicates that the new model enhances performance metrics while reducing computational complexity and model size, making it suitable for deployment on mobile and embedded devices to facilitate rapid development in the apple industry. *Liu & Li (2024)* and *Liu et al. (2024b)* proposed an efficient detection model for apple leaf spot disease, incorporating the Wise-Intersection over Union (IoU) loss function and RepVGG module, which significantly boosted both detection speed and accuracy. Experiments demonstrated that the average accuracy of the improved model was 92.7%, surpassing that of other models and contributing to reduced pesticide usage and improved fruit yield.

With the continuous progress of science and technology, new detection methods are constantly emerging. *Dong et al. (2025)* proposed a Huanglong disease (HLB) non-destructive detection method based on hyperspectral leaf reflectance, which can accurately distinguish the two types of images, but the stability is insufficient under complex light conditions in the field. *Dai et al. (2024)* developed an enhanced lightweight network for high-precision identification of citrus diseases and nutritional deficiencies. Although they have made progress in network structure compression, they have not fully considered multi-level feature interaction and local-global information fusion, resulting in limited generalization ability in small sample disease scenarios. *Thomse et al. (2025)* based on large-scale pre-trained network artificial intelligence algorithm can detect sweet orange green disease, but the model has a large number of parameters and the reasoning speed is less than 20 FPS, which is difficult to meet the real-time requirements in the field. At the feature extraction level, *Huangfu et al. (2024)* proposed the citrus green leaf disease detection method. Although it enhances the global perception ability, it lacks the focusing mechanism for local key areas such as leaf lesions and fruit discoloration. The detection accuracy of small disease features is limited to less than 85%. It is worth noting that *Khuimphukhieo et al. (2024)* evaluated the severity of HLB through uncrewed aerial vehicles (UAV) remote sensing and machine learning, and innovatively combined canopy parameter analysis. However, its method is limited by the professional requirements of UAV control and the adaptability of orchard terrain, which is difficult to achieve large-scale application. *Hridoy et al. (2021)* proposed a modified convolutional neural network, betel leaf CNN (BLCNN), to achieve highly accurate real-time identification of early leaf diseases in betel nut by means of depth-wise separable convolution with Swish activation function technique. Experiments show that the model can effectively control the spread of leaf and root rot diseases, providing an intelligent solution for betel nut plantation industry to reduce economic losses and enhance production capacity.

The above studies have been applied to the field of agricultural engineering. However, most of these models are old and have limited accuracy, and there are three significant defects: First, the lightweight model ignores multi-scale feature interaction; second, high-performance models are accompanied by severe computational redundancy; thirdly, the existing methods are not sensitive enough to small lesions. In addition, due to the relatively single data acquisition environment, it is impossible to perform multi-scenario evaluation on the trained model. Automatic recognition of plant disease images based on deep learning is highly dependent on huge data sets. Therefore, how to improve the detection efficiency of citrus Huanglong disease and reduce the detection cost has become a key problem to be solved in the citrus industry.

Collectively, these studies underscore significant advancements applied in the field of agricultural engineering; However, most existing models for citrus disease identification are outdated and possess limited accuracy, particularly when tested on a small number of samples. While larger network models can improve accuracy to some extent, they also demand significant computational resources. Additionally, the relatively homogeneous environment in which data is collected limits the ability to conduct multi-scene evaluations of the trained models. The automatic recognition of plant disease images through deep

learning heavily relies on large datasets, making it essential to enhance the detection efficiency and reduce the costs associated with identifying Huanglong disease in citrus crops. To address these challenges, this study builds on prior research and incorporates the unique characteristics of citrus diseases to propose targeted enhancements to the YOLOv8 model, resulting in an efficient citrus disease detection algorithm centered on three-channel aggregated attention. The primary contributions of this research are as follows:

(1) Recognizing that most current models for citrus disease detection are outdated, this study adopts and refines the YOLOv8 model to align better with the contemporary requirements of citrus disease detection;

(2) Aiming at the problems of small target feature loss and insufficient multi-scale feature interaction in YOLOv8 in agricultural disease scenes, the Attention attention mechanism is skillfully integrated in the character to float (C2f) structure of the Backbone part to enhance the extraction ability of citrus disease features. At the same time, a lightweight inverse residual moving block (IRMB) module is introduced, which can not only reduce the number of parameters, but also integrate local feature capture and global context understanding, thus significantly improving the performance of the model. This improved structure is named C2f Attention IRMB. In addition, in the Neck part, the Powerneck module is integrated. The module consists of three sub-modules: SimFusion_4in, information fusion module (IFM), and Power-channel. It can not only promote the interactive fusion between different levels of features, but also reduce the amount of redundancy and repeated calculations, aiming to improve the speed and accuracy of model detection.

(3) In response to the prevalent reliance on small sample sizes, single scales, and limited disease types in current citrus disease research, this study develops a comprehensive dataset for citrus Huanglong disease in a natural environment, utilizing data sourced from the Scientific Data Bank, thereby facilitating broader and more in-depth investigations into citrus diseases.

## Data set preparation

The dataset utilized in this study was sourced from the Scientific Data Bank (https://www.scidb.cn/detail?dataSetId=f0d6be4991d846d4874be31cef1ce23f&version=V2). The devices used for photographing were smartphones such as Xiaomi MI9, Huawei and Apple, and the camera was a Sony-RX100 digital camera. The photo distance between the camera and the sample was 50–150 mm,which comprises images captured in natural environments across 12 citrus plantations located in Minhou County, Gutian County, Fuan City, Shunchang County, Meilei District of Sanming, Yong'an City, Youxi County, and Changtai District of Zhangzhou, and the data were collected from 2018 to 2021, Fujian Province. The collection featured a rich diversity of citrus species, including rutabaga (*Citrus reticulate* Blanc Ponkan), banana mandarin (*C. reticulata* var. tankan), honeydew mandarin (*C. unshiu* Marc.), wokan (Orah), red beauty (Hongmeiren citrus hybrid), sugar orange (*C. reticulata* cv. Shatangju), navel orange (*C. sinensis* Osb. var. brasiliensis Tanaka), and shaddock (*C. grandis* var. shatinyu Hort). Sampling occurred under various

weather conditions, encompassing sunny, rainy, and cloudy days, with both white and black panels employed as backgrounds to enhance the diversity of the samples collected (*Chi et al., 2023*). All images were formatted in JPG, with a uniform resolution of 72 pixels per inch and scaled down proportionally to a long side measurement of 640 pixels, capturing both leaves and fruits of citrus. The dataset contained a total of 5,080 images for the training set and 630 images for the test set, representing 12 common symptoms of citrus Huanglong disease, alongside two healthy states: healthy fruit and healthy leaves. The symptoms included blotchy mottling, red-nosed fruit, zinc deficiency, vein yellowing, uniform yellowing, magnesium deficiency, boron deficiency, anthracnose, greasy spot, citrus moss, sooty mold, and canker. Detailed information is provided in Table 1.

## Improved YOLOv8-based algorithms for citrus Huanglong disease field symptom detection

### Overview of the YOLOv8 algorithm

YOLOv8 can be used for real-time object detection, image classification and instance segmentation (*Wang & Wang, 2024*; *Yue et al., 2023*). As a single-stage detector, YOLOv8 integrates image classification, target detection and instance segmentation functions through an end-to-end architecture, which significantly reduces computational redundancy. Its network structure consists of four parts: input image, backbone feature extraction part, neck network and detection head, as shown in Fig. 1. YOLOv8 provides five different size models of N/S/M/L/X scale based on scaling factor to meet the needs of different scenarios. The backbone network and Neck part adopt C2f structure. The Head part uses the current mainstream Decoupled-Head structure to separate the classification and detection heads to avoid task conflicts. Anchor-Free is used to directly predict the center point offset and aspect ratio of the target to reduce the complexity of hyperparameter tuning. In the aspect of loss calculation, the TaskAlignedAssigner positive sample allocation strategy is adopted, and the Distribution Focal Loss is introduced. The data enhancement part of the training introduces the last 10 epochs in YOLOX to close the Mosiac enhancement operation, which can effectively improve the accuracy and inherit the advantages of the previous generation model.

### Improved YOLOv8s module

Leveraging the advantages of the YOLOv8 algorithm, this article proposes a refined and lightweight algorithm for the detection and recognition of citrus Huanglong disease. This enhanced algorithm significantly boosts both accuracy and detection speed when identifying citrus Huanglong disease infestations in complex backgrounds, all while maintaining real-time performance without increasing model complexity. Figure 2 provides a detailed overview of the three-channel aggregated attention citrus disease detection model presented in this study.

### C2f Attention IRMB module

Aiming at the problem of tiny target leakage and feature degradation in the detection of YOLOv8 core network, a C2f module optimization strategy integrating the attention mechanism and the IRMB is proposed. The iso-channel variation strategy and contact

**Table 1 Citrus dataset.** Image credit: Meixing Chi, Shaoping Chen, Ting Huang, Shixiong Chen, Yong Liang, Rongzhou Qiu (https://doi.org/10.57760/sciencedb.j00001.00947). The bold text indicates the citrus Huanglong disease categories as well as the health status, respectively, where there are two healthy categories and 12 disease categories, totalling 5,080 pictures, while sample pictures and corresponding feature descriptions are given for each category.

| Serial number | Citrus Huanglong disease categories and health status | Number of images/sheets | Sample images | Characterization |
|---|---|---|---|---|
| 1 | Healthy fruits | 537 |  | Bright color, solid texture, no pests and diseases, fresh smell |
| 2 | Healthy leaf | 471 |  | Bright green color, no signs of disease spots or insect damage |
| 3 | Blotchy mottling | 586 |  | On newer leaves, the leaf texture becomes hard and brittle, appearing mottled yellow-green |
| 4 | Red nose fruit | 208 |  | Carotenoid accumulation in the pericarp leads to |
| 5 | Zinc-deficiency | 724 |  | Loss of green between leaf veins, leaf size and deformation, spots on leaf blades, scorching of leaf tips and leaf margins; fruit size, spots on the fruit surface |
| 6 | Vein-yellowing | 391 |  | Yellowing of leaves, usually starting with the veins first |

(Continued)

| Serial number | Citrus Huanglong disease categories and health status | Number of images/sheets | Sample images | Characterization |
|---|---|---|---|---|
| 7 | Uniform yellowing | **275** |  | It is caused by a gram-negative bacterium on the newest leaves, causing a uniform yellowing of the entire leaf, and can result in the destruction of an entire orchard of citrus trees. |
| 8 | Magnesium-deficiency | 262 |  | Loss of green between the veins, the leaf veins remain green, the leaf blade as a whole shows yellow-green mottled, there may also be growth retardation, fruit size, fruit quality degradation and other symptoms |
| 9 | Boron-deficiency | 152 |  | New and old leaves may be scorched, deformed, or yellowed, with dead spots on the leaf margins, as well as brittle leaves, small fruits, rough skin, and possible deformities. |
| 10 | Anthracnose | 286 |  | Water-soaked dark spots appear on the leaves, which then gradually expand and become round or irregular, with a black or dark brown center and a yellow halo around the edges. |
| 11 | Citrus greasy spot | 202 |  | Usually on the abaxial surface of the leaf, initially the abaxial surface of the leaf will appear the size of a pinhead of discoloration spots, these spots from translucent to bright view will gradually expand into yellow patches |
| 12 | Citrus moss | 298 |  | Leaves may take on an unnatural green or gray color, due to the moss cover that prevents the leaves from photosynthesizing properly |

| Serial number | Citrus Huanglong disease categories and health status | Number of images/sheets | Sample images | Characterization |
|---|---|---|---|---|
| 13 | Sooty mold | 327 | | Leaves may take on an unnatural green or gray coloration |
| 14 | Canker | 361 | | Initially, pinhead-sized yellow greasy spots appeared on the leaf abaxial surface, enlarged and rounded, with a central bulge and, a corky, rough, and crater-like surface. There is a yellow halo around the spots. |
| Total | | 5,080 | | |

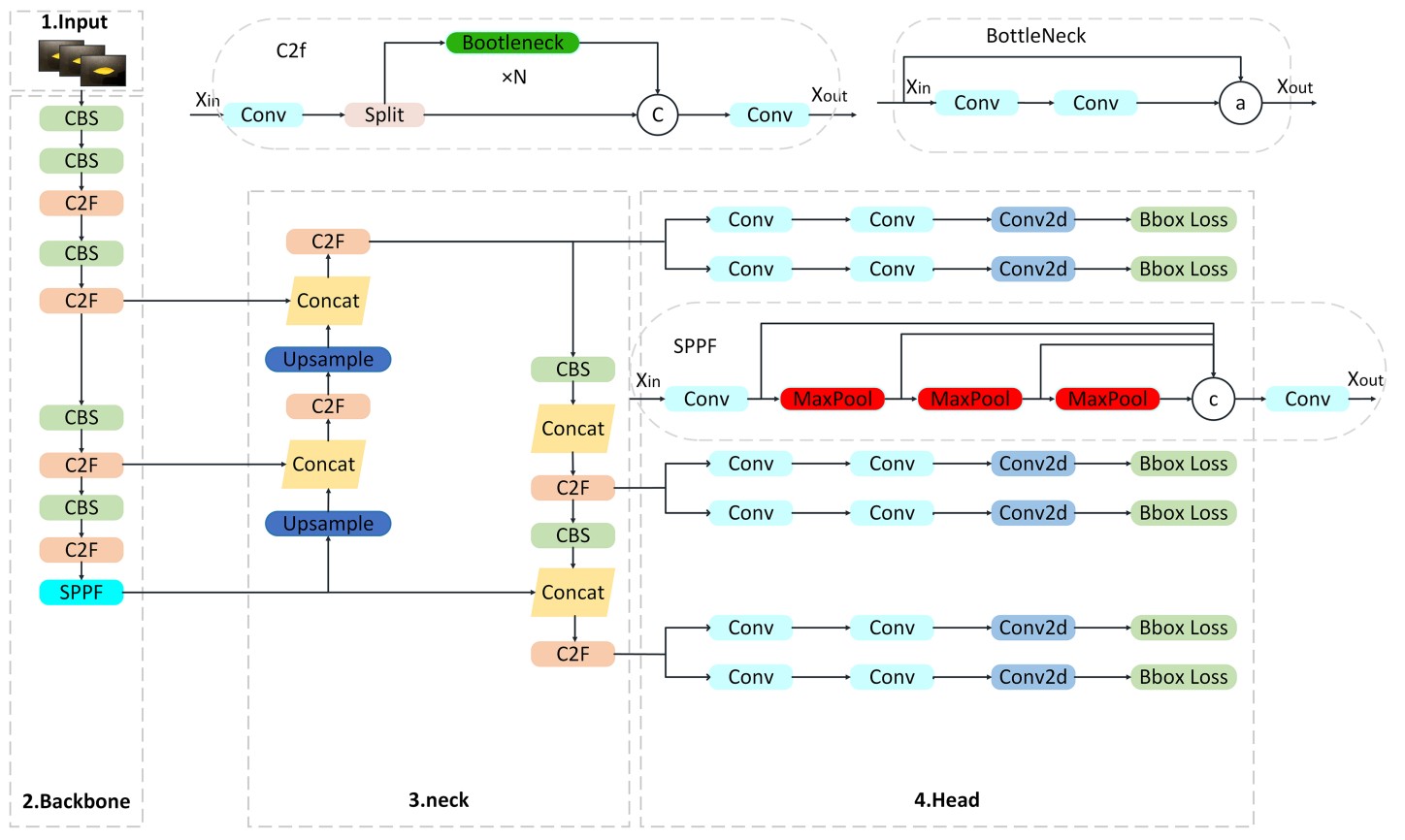

**Figure 1** YOLOv8 network architecture diagram.

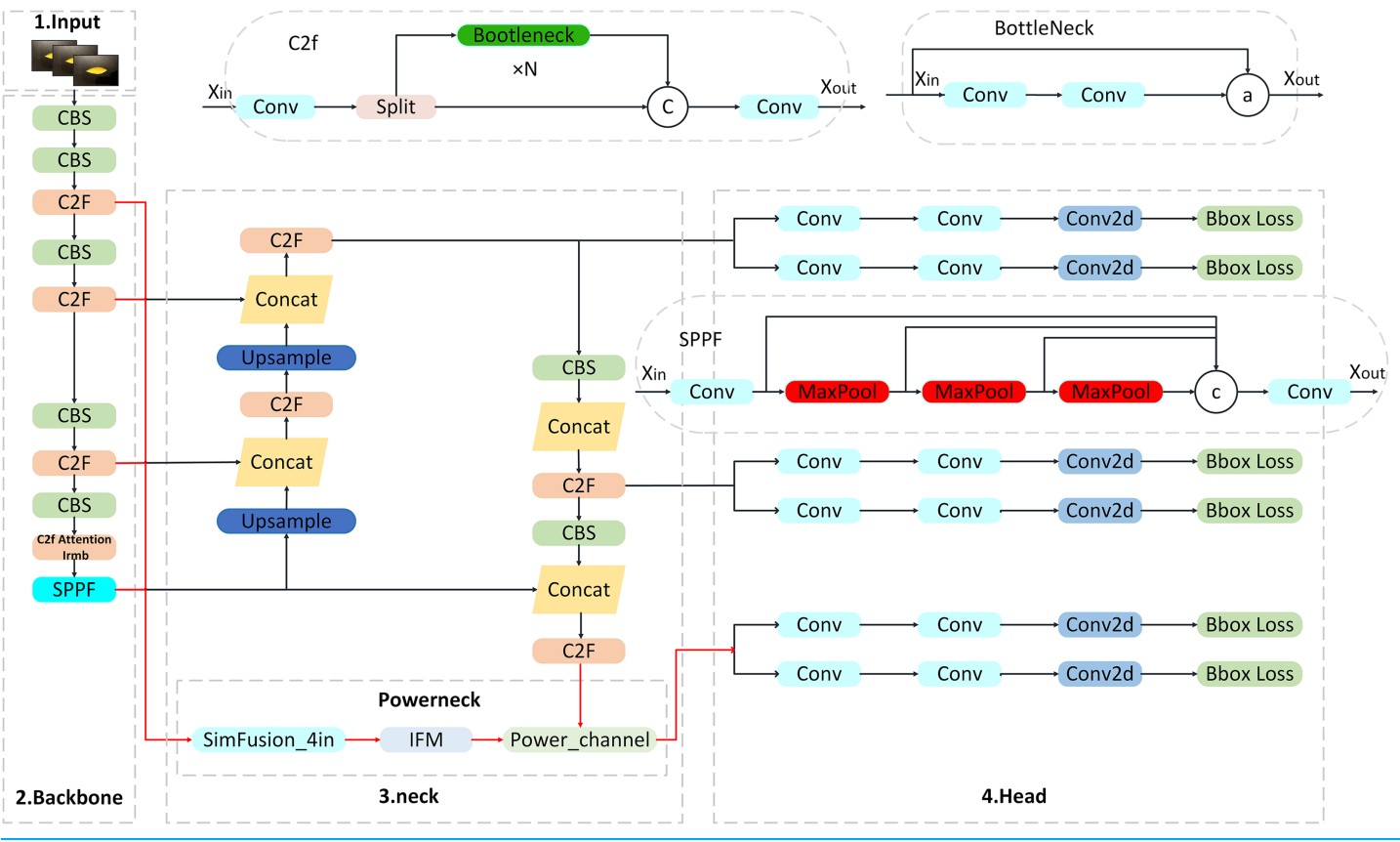

**Figure 2 Structure of three-channel aggregated attention network.**

downsampling of the original C2f module lead to a sudden drop in the resolution of the deep feature map (the resolution is lower than 20 × 20 when the input is 640 × 640), the effective pixel ratio of tiny targets (*e.g.*, spots at the early stage of the lesion) is less than 1.4% (*Lin et al., 2017*), and the key texture information is seriously lost in the process of spatial compression, and moreover the fixed channel dimensionality strategy in the standard Bottlenneck unit limits the multiscale detection and feature degradation problems. Channel dimension strategy in the standard Bottleneck cell limits the ability of multi-scale feature characterization (*Zhang et al., 2024b*). To solve these problems, in this article, by integrating the attention mechanism in the original C2f module and adopting the grouped fully connected optimization module in its channel branch, the computational volume is reduced while the small target feature response is enhanced, and secondly, the IRMB module is introduced in the Context-Based Spatial Attention Mechanism (CBS) (S = 1, K = 1) module at the end of the module to strengthen the local feature reuse efficiency through dynamic deep convolution and channel mixing and shuffling operations (*Chen et al., 2020*), and the synergy between the attention mechanism and IRMB synergistically form a closed-loop feature recalibration mechanism, and the high-level semantic feedback from IRMB will dynamically update the front-level attention weights to

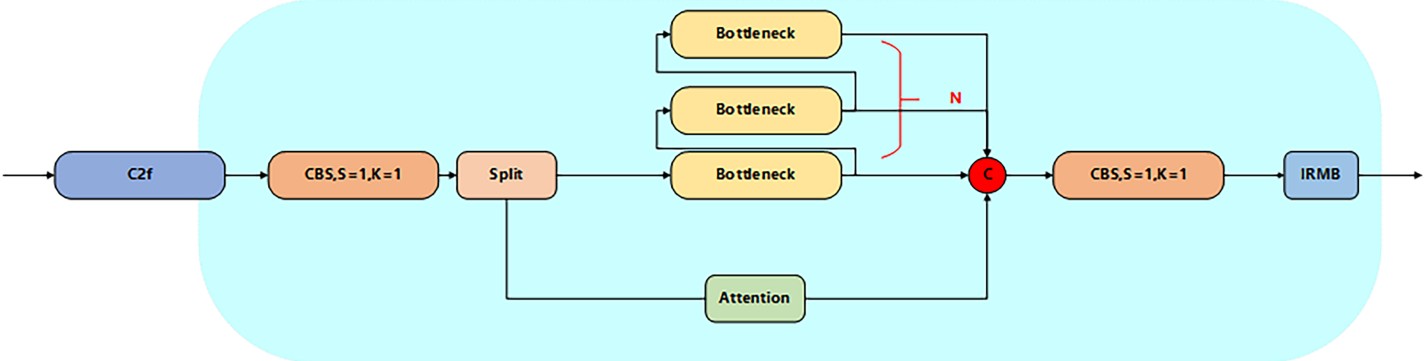

**Figure 3 Structure of C2f Attention IRMB.**

significantly optimize the feature expression capability and thus improve the accuracy, and the improved module is shown in Fig. 3.

### Powerneck module

The Powerneck module consists of three modules: SimFusion_4in, IFM and Power_channel, which form a closed-loop optimization system through a cascade-feedback mechanism. SimFusion_4in firstly aligns the shallow high-resolution features and preserves the microstructure of leaf lesions; IFM, based on this, fuses the global context to establish the spatial correlation between lesion areas and healthy tissues; and Power_channel strengthens the cross-layer transfer efficiency of key features and suppresses background noise interference through dynamic channel brushing. The three achieve synergy through feature pyramid backpropagation: the higher-order semantic features output from Power_channel are fed back to SimFusion_4in, which triggers adaptive pooling kernel size adjustment to form a closed loop of multi-scale perception, thus improving the accuracy, as described below.

### SimFusion_4in module

SimFusion_4in, also known as Low-stage Feature Alignment Module (LoW-FAM), optimizes shallow feature representation through a multi-scale feature fusion strategy, as shown in Fig. 4. The module first uses multi-stage average pooling to downscale the input features, and uniform up/down-sampling expansion to ensure consistent dimensions, in order to eliminate redundant information and retain key texture features. Subsequently, cross-channel splicing is used to realize feature aggregation to ensure the accuracy of the information and also reduce the complexity of the operation. In the feature alignment stage, a lightweight attention mechanism is introduced to achieve cross-layer feature calibration through Softmax weight assignment of channel dimensions, and RepBlock (*Ding et al., 2021*) is embedded at the end of the attention output, which uses structural reparametrization to merge the $3 \times 3$ convolution in the training stage with the $1 \times 1$ convolution in the inference stage to further improve the feature extraction efficiency, and deepen the extraction and integration of information. And there are also related experiments showing that LoW-FAM can enhance the edge response strength of shallow

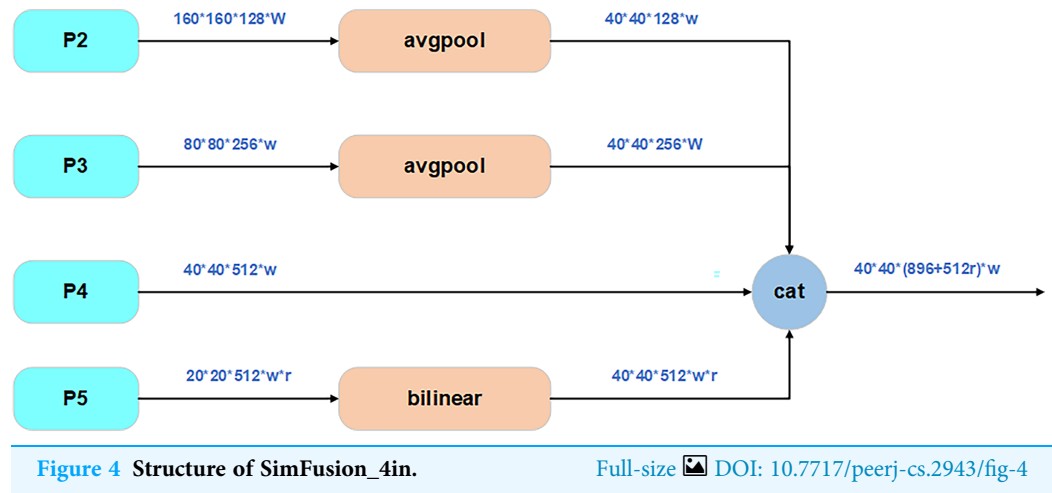

**Figure 4** Structure of SimFusion_4in.               

features and significantly improve the localization accuracy of tiny lesion areas (*Liu et al., 2024a*).

### IFM module

IFM focuses on dynamic integration with global semantic information, as shown in Fig. 5. The module adopts a dual-path structure: the primary path captures the Great Wall dependencies by cavity convolution (dilation = 2), while the secondary path extracts local detail features by group convolution. The two-path features are fused by adaptive weighting to generate a global context descriptor with dual spatial-channel perception. In order to strengthen the feature distribution efficiency, a layered injection mechanism is designed: the Global Context Descriptor (GCD) is decomposed into multi-scale components by deformable convolution, and gated attention is dynamically distributed to different network layers. And the validation on the VisDrone dataset shows that IFM can effectively improve the detection recall of dense lesion regions, while reducing the duplicate detection frames and improving the detection of small targets (*Xiong et al., 2025*).

### Power_channel module

Power_channel module as a dynamic regulator of feature transfer, with a weighted role, through the resource-aware mechanism to optimize cross-layer feature interaction, can enhance the focusing ability of the anchor frame, to achieve the average accuracy as well as performance improvements. The module consists of five parts that comprise the core structure of the five phase processing: convolution (Conv), reparameterized convolution (RepConv), the hybrid pooling layer (Pool), the fully-connected (FC) layer, and the activation function (ACT), as shown in Fig. 6. The process is divided into five parts: channel compression, feature reconstruction, context aggregation, dynamic weight generation, and feature recalibration. Firstly, the input channel is downscaled by $1 \times 1$ convolution to reduce computational redundancy, secondly, the reparameterized convolution fuses $3 \times 3$ and $1 \times 1$ convolution to enhance the spatial characterization ability, then the hybrid pooling layer performs the maximal pooling and average pooling in parallel to generate the multi-granularity statistical features, next, the fully-connected layer and the activation

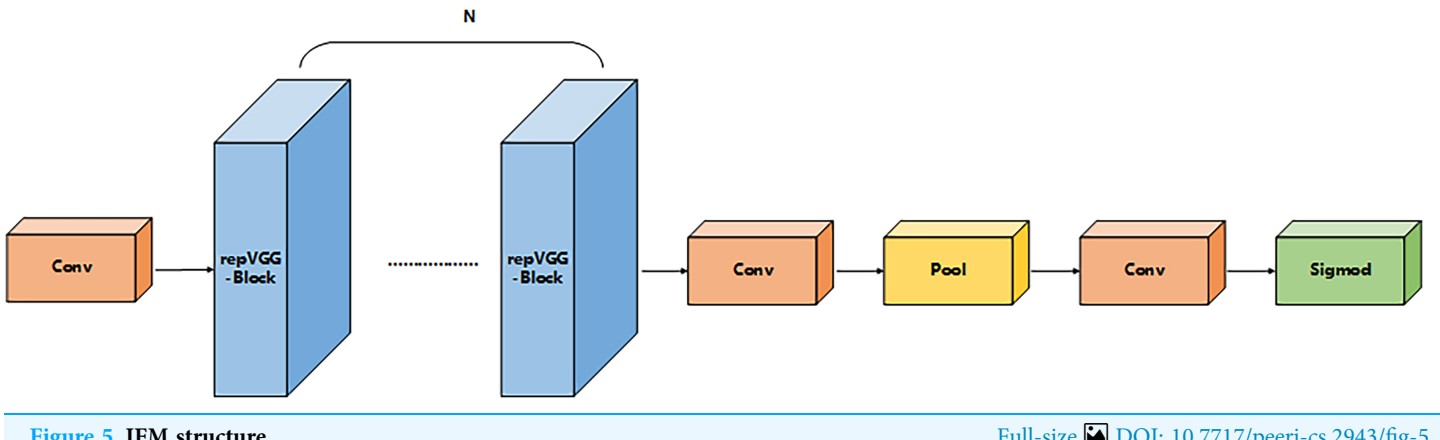

**Figure 5 IFM structure.**

**Figure 6 Power_channel structure.**

function jointly compute the channel attentional weights, and finally, the weights are multiplied with the original feature channel to achieve the focused feature enhancement. Through such a union, Power_channel efficiently passes features between layers, ensures that each part receives the information it needs, and dynamically adjusts resource allocation according to the computational needs of different layers to reduce redundancy and duplicate computation, thus improving the efficiency and performance of the overall model.

### Pseudo-code for improved modules

The pseudo-code corresponding to the C2f Attention IRMB module is as follows:

Class C2f_Attention_1RMB inherits from C2f:

    Constructor(c1, c2, n=1, shortcut=False, g=1, e=0.5):

        Call parent class constructor

        Initialize module list 'm' with n instances of BottleneckByAttention:

            Each instance has parameters (self.c, self.c, shortcut, g, k=((3,3), (3,3)), e=1.0)

        Define conv layer cv2 with input channels (2+n)*c and output channels c2

        Initialize CD_LSK module 'cd' with input c and output 2*c

    Forward pass(x):

        Split input x through cv1 into two parts x1 and x2

        Initialize list y with:

           cd(x1)

x2

Iteratively process through modules in 'm':

For each module in self.m:

Apply module to last element of y

Append result to y

Concatenate all elements in y along channel dimension (dim=1)

Pass concatenated result through cv2

Return final output

The pseudo-code corresponding to the SimFusion_4in module is as follows:

Class SimFusion_4in inherits from nn.Module:

Constructor():

Initialize adaptive average pooling function reference

Forward(x):

Unpack four input tensors (x_l, x_m, x_s, x_n)

Get spatial dimensions from x_s

Prepare output_size = [H, W]

Handle ONNX export special case:

Replace avg_pool with ONNX-compatible version

Process inputs:

x_l = adaptive_avg_pool2d(x_l, output_size)

x_m = adaptive_avg_pool2d(x_m, output_size)

x_n = bilinear_interpolate(x_n to output_size)

Concatenate all processed tensors along channel dimension

Return fused feature tensor

The pseudo-code corresponding to the IFM module is as follows:

Class IFM inherits from nn.Module:

Constructor(inc, ouc, embed_dim_p, fuse_block_num=3):

Initialize sequential convolution layers:

Conv layer from inc to embed_dim_p

Repeated RepVGGBLock blocks (count=fuse_block_num)

Final Conv layer from embed_dim_p to ouc

Initialize adaptive average pooling (output size 1x1)

Initialize 1x1 convolution fc layer (ouc->ouc)

Initialize Sigmoid activation

Forward(x):

Process input through conv layers

Apply adaptive pooling

Pass through fc layer

Apply Sigmoid activation

Return activated feature weights

The pseudo-code corresponding to the power_channel module is as follows:

Class power_channel inherits from nn.Module:

Constructor():

No parameters needed

Forward(x):

Return element-wise product of x[0] and x[1]

## Model training

### Experimental platforms

The experiments were conducted on a server equipped with 120GB of RAM, an NVIDIA GeForce RTX 4090 GPU, and a 12-core CPU (*Liu, Yu & Geng, 2024*), detailed specifications are provided in Table 2. The training operations were implemented in Python using the PyCharm IDE (*Jin et al., 2024*), with the parameters set to 200 epochs, a batch size of 64, works of 4, IoU of 0.5, and a learning rate of 0.0001.

### Methodology for assessing indicators

Upon completing the training, the model underwent evaluation using precision, recall, accuracy, and F1 score as metrics. For the multi-classification task, confusion matrices were employed as the primary evaluation criterion, and the classification results are presented in Table 3, where true positive (TP), false positive (FP), true negative (TN), and false negative (FN) represent the true positive, false positive, true negative, and false negative instances, respectively, and the total number of samples is represented by the equation TP + FP + TN + FN = Total samples (*Sun, Feng & Chen, 2024*; *Subha & Kasturi, 2024*).

In Table 3, TP denotes the number of positive classes predicted as positive, which is the number of positive class samples that the classifier recognizes as correctly predicted; FN denotes the number of positive classes predicted as negative, which is the number of positive class samples that the classifier incorrectly predicts; FP denotes the number of negative classes predicted as positive, which is the number of negative class samples that the classifier recognizes the number of negative class samples incorrectly predicted; TN denotes the number of negative classes predicted as negative, which is the number of negative class samples that the classifier considers correctly predicted (*Prasad et al., 2024*).

The precision rate is the proportion of actual positive cases in all samples predicted by the model to be positive cases, and is calculated by the formula:

$$Precision\ (P) = \frac{TP}{TP + FP}. \tag{1}$$

Recall is a measure of how many positive cases are classified as positive in all samples that are actually positive cases, and it measures the model's ability to recognize positive cases, which is calculated as:

$$Recall\ (R) = \frac{TP}{TP + FN}. \tag{2}$$

A higher accuracy measure represents a better model, which is calculated as:

$$Accuracy = \frac{TP + TN}{TP + FN + FP + FN}. \tag{3}$$

**Table 2 Experimental environment.**

| Serial number | Installations | Detailed information |
|---|---|---|
| 1 | CPU | 12 crux |
| 2 | GPU | NVIDIA GeForce RTX 4090 |
| 3 | RAM | 120GB |
| 4 | Cuda version | 11.3 |
| 5 | Python version | Python3.8 |
| 6 | Pytorch framework version | Pytorch 1.11.0 |

**Table 3 Structure of C2f Attention Irmb.** The real situation projected results standard practice counter-example standard practice TP FN counter-example FP FN.

| The real situation | Projected results | |
|---|---|---|
| | Standard practice | Counter-example |
| Standard practice | TP | FN |
| Counter-example | FP | FN |

**Table 4 Comparison of experimental results of the improved model.**

| Models | Precision/% | MAP 50/% |
|---|---|---|
| YOLOv8s | 90.3 | 95.9 |
| YOLOv8s+C2f | 92.7 | 96.8 |
| YOLOv8s+Powerneck | 91.9 | 96.7 |
| YOLOv8s+Powerneck+C2f | 91.5 | 97.0 |

F1 is the average of precision and recall, a measure of the classification problem, with 1 representing the best and 0 the worst.

$$F1 = \frac{2 * Precision * Recall}{Precision + Recall} = \frac{2TP}{2TP + FN + FP}. \tag{4}$$

## Experimental results and analysis

### Analysis of the experimental results of the improved model

In order to verify the synergy between the C2F Attention IRMB (hereafter abbreviated as C2f) and the Powerneck module, this article sets up a set of ablation experiments to comparatively analyze the performance of the two models (*Liu et al., 2021*).

(1) The original model YOLOv8s; (2) the YOLOv8s model with the introduction of the C2f module; (3) the addition of the Powerneck module; and (4) the addition of the modules mentioned in (2) and (3). The citrus dataset was trained in the same experimental environment and the results are shown in Table 4. As can be seen from the table, the improved method used in this article significantly improves the performance of the original network, the accuracy is significantly improved, and the MAP50 finally reaches 97%.

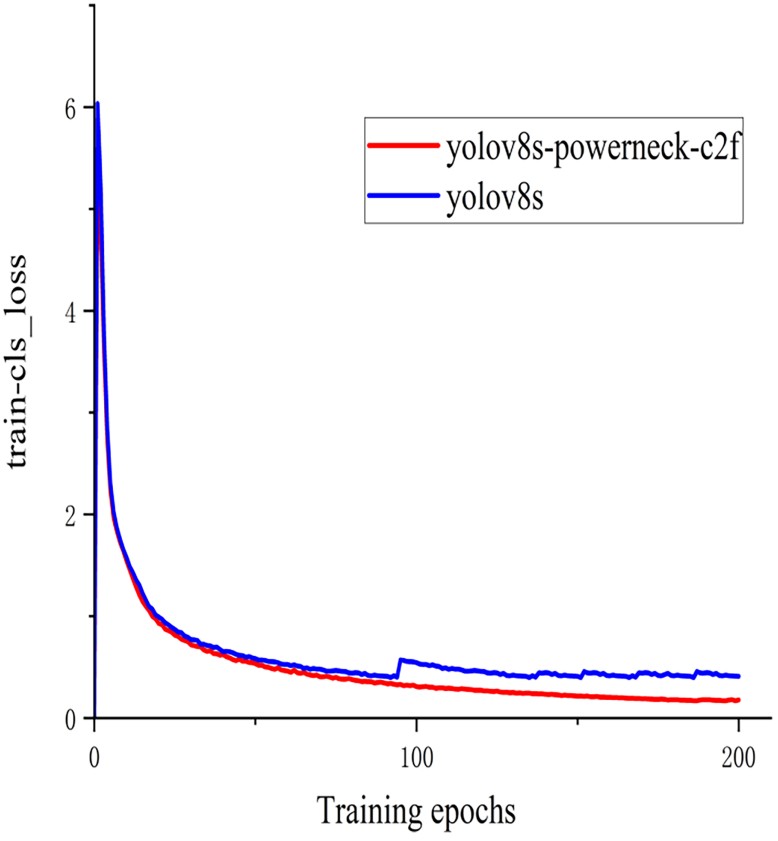

**Figure 7 Graph of training classification loss function.**

In order to better demonstrate this experiment (*Barman & Choudhury, 2022*), the training classification loss value and test bounding box loss value function plots of the model YOLOv8s of the above experiment and the method YOLOv8s-Powerneck-C2f experiment of this article are shown in Figs. 7 and 8. As can be seen from Fig. 7, the training classification loss function converges to 0.12 in 100 epochs, which is 33.3% lower than the original model of 0.18, indicating that the attention mechanism effectively suppresses the background noise interference and accelerates the model to focus on the key features of the lesion region; as can be seen from Fig. 8, the test bounding box loss function stabilizes at 0.4 in 100 epochs, which is 0.45 lower than the original model of 0.45 11.1%, proving that the aligned features can locate tiny lesions more precisely; the results show that the model can effectively reduce the detection loss and improve the detection accuracy.

### Comparative analysis of different models

To further demonstrate the validity of this experiment, comparisons between different models were performed under the same experimental conditions (same environment as well as parameters) (*Yasmeen et al., 2022*). In this article, we use YOLOv5m, YOLOv6s, YOLOv7x, YOLOv8s, YOLOv8-p6, YOLOv8-ghost, and YOLOv8s-powerneck-C2f for the test and comparison, and the experimental results are detailed in Table 5.

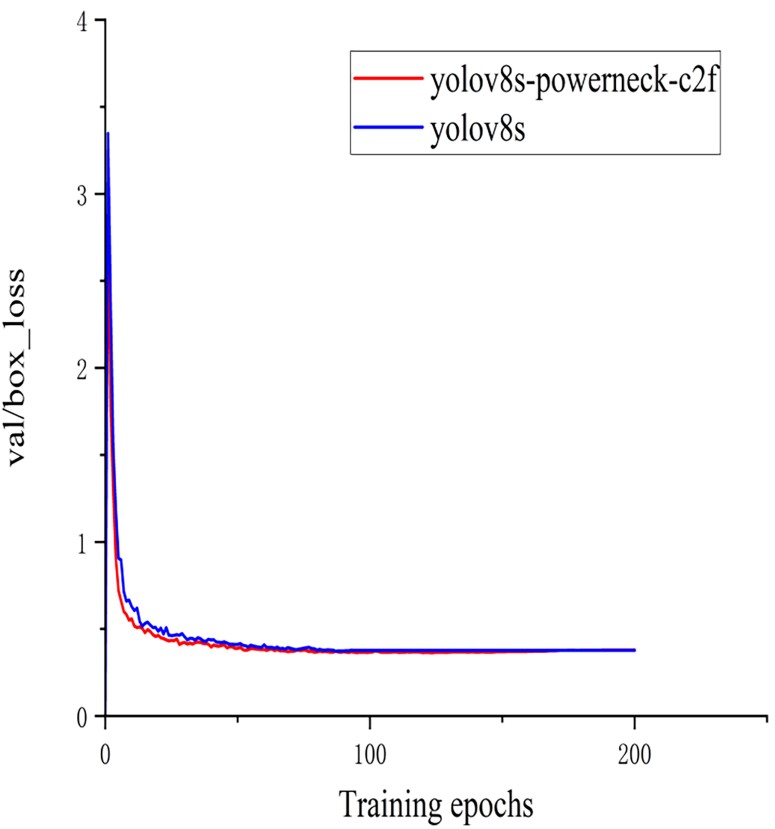

**Figure 8 Plot of test bounding box loss function.**

**Table 5 Comparison of experimental results for different models.**

| 1 | Models | MAP 50/% | MAP50-95/% | FPS |
|---|---|---|---|---|
| 2 | YOLOv5m | 95.9 | 89.8 | 333 |
| 3 | YOLOv6s | 96.5 | 91.4 | 333 |
| 4 | YOLOv7x | 96.7 | 91.4 | 110 |
| 5 | YOLOv8s | 95.9 | 90.4 | 294 |
| 6 | YOLOv8-p6 | 96.5 | 91.2 | 286 |
| 7 | YOLOv8-ghost | 95.9 | 90.4 | 238 |
| 8 | YOLOv8s-powerneck-C2f | 97.0 | 91.4 | 370 |

The map50 for each model is shown in Fig. 9 (*Arthi et al., 2023*). The inference speed is shown in Fig. 10; from Fig. 9, YOLOv8s-powerneck-C2f has faster convergence and higher average accuracy compared with other models; from Fig. 10, YOLOv8s-powerneck-C2f has excellent inference speed and achieves better FPS performance.

### *Example of recognition result display*

Table 6 shows a typical example of the results of this model used for citrus disease identification (*Neves et al., 2023*), which can successfully identify the citrus Huanglong disease in China studied in this research.

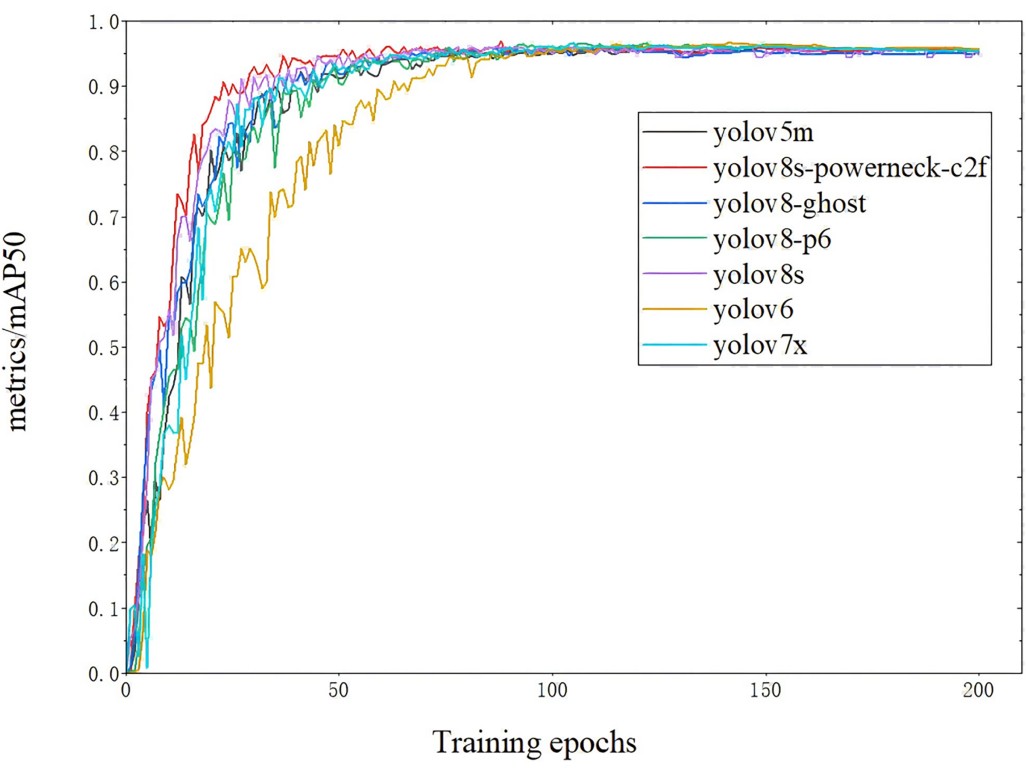

Figure 9 **Comparison of Map50 for each model.**

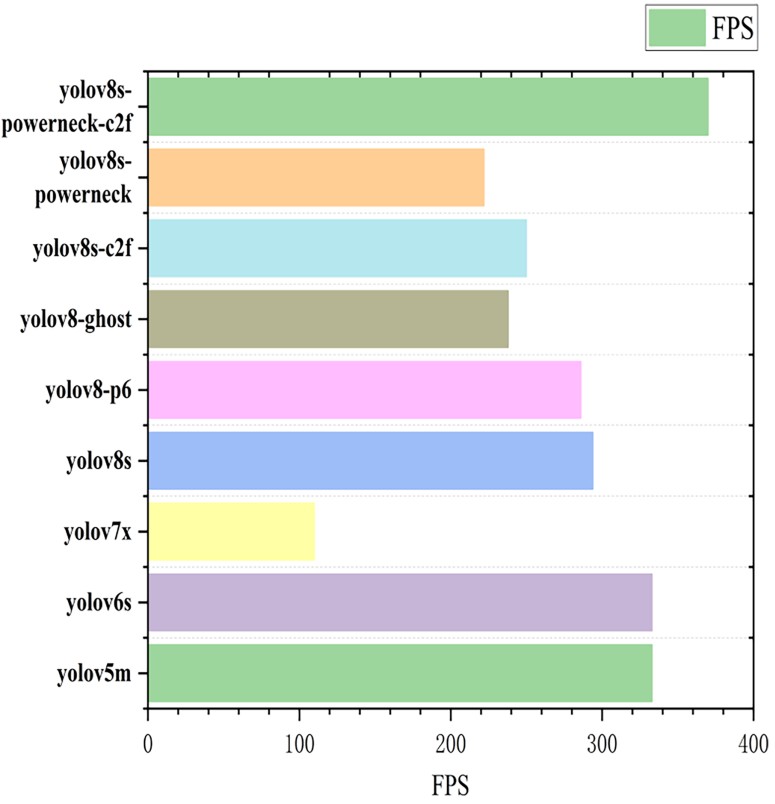

Figure 10 **Plot of inference speed for each model.**

**Table 6 YOLOv8 improved detection result.** Image credit: Meixing Chi, Shaoping Chen, Ting Huang, Shixiong Chen, Yong Liang, Rongzhou Qiu (https://doi.org/10.57760/sciencedb.j00001.00947).

| Citrus Huanglong disease categories and health status | Results 1 | Results 2 |
|---|---|---|
| Blotchy motting |  |  |
| Red nose fruit |  |  |
| Healthy leaf |  |  |
| Healthy fruit |  |  |
| Vein yellowing |  |  |
| Uniform yellowing |  |  |

| Citrus Huanglong disease categories and health status | Results 1 | Results 2 |
|---|---|---|
| Zinc deficiency |  |  |
| Anthracnose |  |  |
| Canker |  |  |
| Soot mould |  |  |
| Magnesium deficiency |  |  |
| Citrus greasy spot |  |  |

## CONCLUSIONS

Aiming at the problems of insufficient feature sensitivity, poor robustness in complex environments, and difficulty in synergistically optimizing the detection accuracy and inference speed that are commonly found in the current citrus disease detection models, this study proposes a YOLOv8s-Powerneck-C2f detection method based on YOLOv8 improvement for efficient identification of citrus Huanglong disease disease symptoms in complex field scenes. The working points of this study can be summarized as follows: (1) lightweight feature extraction with attention synergy: a C2f Attention-IRMB module is designed to replace the traditional C2f layer by dual attention mechanism integration and inverse residual moving block, which reduces the computational volume while improving the detection accuracy; (2) dynamic optimization of multilevel feature fusion: the Powerneck architecture is proposed to replace the traditional C2f layer by means of dynamic channel pruning and hybrid pooling strategies to achieve cross-scale feature fusion efficiency and reduce redundancy and repetitive computation; (3) structural reparameterization acceleration strategy: optimizing the design of the detection head and structural dynamic parameter restructuring technology to increase the inference speed by 14.6% while maintaining the accuracy to meet the real-time field detection. Based on the comprehensive citrus dataset covering 12 disease states and two health categories experiments showed that the improved model achieved effectiveness and superiority, the present model with 97% mAP50 and 91.5% Precision compared to the baseline YOLOV8 improved 1.1% and 1.2%, respectively, which provides an effective detection tool for citrus Huanglong disease field production, and has important theoretical and practical significance. Although this study has achieved some results in citrus disease identification, the following limitations still exist: first, the experimental data are mainly derived from a specific region in Fujian Province, China, and the samples are not sufficiently diverse in terms of climate and soil conditions, which may limit the model's ability to generalize across different geographic environments. Second, the model is highly dependent on high-performance GPUs in the inference process, and further verification of its real-time performance and deployment feasibility on resource-constrained devices is needed. In future research, on the one hand, we will focus on the exploration of cross-region and cross-species migration learning. On the other hand, we will continue to optimize the detection model to improve the detection speed under the premise of guaranteeing the accuracy and promote the scale application of this technology in resource-constrained scenarios.

### Funding

This work was supported by the National College Student Innovation Training Program, Number: 202413558003; the Xinjiang Uygur Autonomous Region undergraduate education teaching research and reform project, Project Number: XJGXZHJG-202347; and sponsored by the Natural Science Foundation of Xinjiang Uygur Autonomous Region

(2022D01C461, 2022D01C460). The funders had no role in study design, data collection and analysis, decision to publish, or preparation of the manuscript.

## Grant Disclosures

The following grant information was disclosed by the authors:
National College Student Innovation Training Program: 202413558003.
Xinjiang Uygur Autonomous Region: XJGXZHJG-202347.
Natural Science Foundation of Xinjiang Uygur Autonomous Region: 2022D01C461, 2022D01C460.

## Competing Interests

The authors declare that they have no competing interests.

## Author Contributions

- Yizong Wang conceived and designed the experiments, performed the experiments, analyzed the data, performed the computation work, prepared figures and/or tables, authored or reviewed drafts of the article, and approved the final draft.
- Zhengrong Xiao conceived and designed the experiments, performed the experiments, analyzed the data, performed the computation work, prepared figures and/or tables, authored or reviewed drafts of the article, and approved the final draft.
- Hong Wang conceived and designed the experiments, performed the experiments, analyzed the data, performed the computation work, prepared figures and/or tables, authored or reviewed drafts of the article, and approved the final draft.
- Fei Li conceived and designed the experiments, performed the experiments, analyzed the data, performed the computation work, prepared figures and/or tables, authored or reviewed drafts of the article, and approved the final draft.
- Jiya Tian conceived and designed the experiments, performed the experiments, analyzed the data, performed the computation work, prepared figures and/or tables, authored or reviewed drafts of the article, and approved the final draft.

## Data Availability

The Image datasets of Citrus Huanglongbing field symptom recognition are available at Chi Meixiang, Chen Shaoping, Huang Ting, Chen Shixiong, Liang Yong, and Qiu Rongzhou. 2024. "Image Datasets of Citrus Huanglongbing Field Symptom Recognition." Science Data Bank. doi: 10.57760/sciencedb.j00001.00947.

## Supplemental Information

Supplemental information for this article can be found online at http://dx.doi.org/10.7717/peerj-cs.2943#supplemental-information.

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
