# Peer review of "A study on an efficient citrus Huanglong disease detection algorithm based on three-channel aggregated attention"

_PeerJ Computer Science, doi:10.7717/peerj-cs.2943_

## Round 0.1 · original submission · Major Revisions

After a careful analysis of the manuscript, I believe that further work is needed to improve its clarity and completeness. In particular, it is essential to motivate in more detail the changes made to YOLOv8 and provide a more in-depth comparison with existing methodologies, highlighting the research gap.

Furthermore, the paper would benefit from a strengthening of the experimental section with additional illustrations and a clearer presentation of the algorithm flow, possibly with a pseudo-algorithm to facilitate its understanding.

From a structural point of view, the abstract needs a greater emphasis on the results and main conclusions of the study. The introduction should be expanded with more contextual details, updated references and a clear exposition of the contributions of the work. The dataset and the image preparation process should also be described more precisely. The discussion of the algorithms should be more in-depth, explaining in detail their flow and the modules involved. Similarly, the training process requires a more detailed explanation of the indicators used, supported by relevant references.

The analysis of the results should be expanded with a more detailed discussion of the main findings of the study. The concluding section should also be reworded to ensure greater clarity and completeness. Finally, careful linguistic revision is necessary to improve the fluency of the text and correct any inaccuracies, as well as careful checking of graphs, tables and equations to ensure their coherence and accuracy. In light of these aspects, a major revision is necessary before the manuscript can be considered for acceptance.

Reviewer 1 ·

Basic reporting

1. Article presented for the journal aims to present study on efficient citrus Huanglong disease detection using improved Yolov8 and related study analysis.
2. Organizational structure of the paper and illustrations is fine.
3. The most significant topic of research is the modified Yolov8 as choice of study. Authors need to justify, why improvement is made in Yolov8.
4. Dataset is chosen which is open access which is good for other research group and defend their study.
5. Results, analysis is fine. More experimental illustrations is needed.
6. Summary of existing approach is needed which would clearly highlight the exiting work and research gap.
7. Pseudo algorithm could have been presented in the paper for better understanding of the algorithm.
8. References cited in the paper is fine.

Experimental design

The experimental illustrations, flow and comparative results with sample images (input and corresponding output) is acceptable.

Validity of the findings

The findings are well validated by comparing with other existing approaches by smaller scale and it is recommended to researchers to present depth analysis as compared with other approaches.

Additional comments

The paper needs technical depth as far as model description is concerned.

·

Basic reporting

1. The Abstract section needs improvement; include sufficient details about the outcomes and findings of this study.
2. Revise the Introduction section to add more background information, highlight the importance of the research, and clearly state the focus of this study.
3. Add more recent related articles in the Introduction section.
4. Clearly and concisely outline the primary contributions of this research at the end of the Introduction section.
5. In the Dataset Preparation section, provide more comprehensive details about the dataset and the images used in the study.
6. In the Improved YOLOv8-Based Algorithms for Citrus Huanglong Disease Field Symptom Detection section, elaborate on the algorithms and modules. Explain the workflow of all algorithms and modules in detail.
7. In the Model Training section, describe all indicators thoroughly and include appropriate references to support the explanation.
8. In the Experimental Results and Analysis section, provide a detailed discussion of the findings and outcomes of this research.
9. Revise the Conclusion section to ensure clarity and completeness.
10. Perform a strong proofreading of the entire paper.
11. Revise the entire paper to address the grammatical issues identified in the submitted manuscript.
12. Carefully recheck all graphs, tables, and equations for accuracy and consistency.

Experimental design

no comment

Validity of the findings

no comment

Additional comments

no comment

---

## Round 0.2 · Minor Revisions

The manuscript addresses an interesting and timely topic, but requires some revisions before being considered for publication. Greater clarity is requested in the Discussion section of the abstract and a strengthening of the literature review, including recent studies such as Hridoy et al. (2021). It is also necessary to provide more details about the dataset, in particular about the image acquisition conditions. Punctuation and spacing errors are also reported (e.g. lines 375–376), to be corrected with a careful proofreading. Finally, it is suggested to add a brief discussion on the limitations of the study in the Conclusion section.

·

Basic reporting

The article contains several punctuation errors.

Experimental design

no comment

Validity of the findings

no comment

---

## Round 0.3 · accepted · Accept

The document is ready for publication. I just recommend including a comparison table that contrasts this study with existing literature.

Reviewer 1 ·

Basic reporting

Paper is well drafted and the article is clear and concise.

Experimental design

The design and analysis of proposed system is clear and drafting is good.

Validity of the findings

The findings of the study and illustrations is good and presented in the paper in tables and graphs.

Additional comments

Revised paper is good.

·

Basic reporting

In the Experimental Results and Analysis section, include a comparison table that contrasts this study with existing literature. This will clearly highlight the novelty and outcomes of the current work.

Experimental design

no comment

Validity of the findings

no comment

Additional comments

no comment